# PREventing Mild Idiopathic SCOliosis PROgression (PREMISCOPRO): A protocol for a randomized controlled trial comparing scoliosis-specific exercises with observation in mild idiopathic scoliosis

Elias Diarbakerli[1,2]*, Allan Abbott[3,4], Paul Gerdhem[1,2,5]

1 Department of Clinical Science, Intervention and Technology, Karolinska Institutet, Stockholm, Sweden, 2 Department of Reconstructive Orthopaedics, Karolinska University Hospital, Stockholm, Sweden, 3 Division of Prevention, Rehabilitation and Community Medicine, Department of Health, Medicine and Caring Sciences, Unit of Physiotherapy, Linköping University, Linköping, Sweden, 4 Department of Orthopaedics, Linköping University Hospital, Linköping, Sweden, 5 Department of Orthopaedics, Department of Surgical Sciences, Uppsala University Hospital, Uppsala University, Uppsala, Sweden

* elias.diarbakerli@regionstockholm.se

**Data Availability Statement:** No datasets were generated or analysed during the current study. All

## Abstract

### Background

Idiopathic scoliosis is the most common spinal deformity in children. Treatment strategies aim to halt progression of the curve. Mild scoliosis is in many cases observed or, in some cases, treated with scoliosis-specific exercises. More severe curves are treated mainly with a brace. The aim of this study is to investigate the effectiveness of scoliosis-specific exercises compared to observation in adolescents with mild idiopathic scoliosis.

### Methods

**Subjects**. Previously untreated and skeletally immature children aged 9–15 years of age with idiopathic scoliosis (curve magnitude Cobb 15–24 degrees) will be included. A total of 90 subjects will be included to receive one of two possible interventions.

**Interventions**. Both groups will receive a physical activity prescription according to the World Health Organization recommendations. The intervention group will receive an additional active self-correction treatment strategy for curve correction and will have outpatient sessions once every two weeks for the first three months. They will be prescribed to do the exercises at least three times per week. The intervention will be performed until skeletal maturity or progression of the curve.

**Outcome**. The subjects will participate in the study until curve progression or until skeletal maturity (defined as less than 1 cm growth for six months). The primary outcome variable is failure of treatment, defined as progression of the Cobb angle more than 6 degrees on two consecutive x-rays compared to the baseline x-ray. Secondary outcome measures include patient-reported outcomes, clinical characteristics (i.e. angle of trunk rotation and trunk

relevant data from this study will be made available upon study completion.

**Funding:** This study has been funded by Region Stockholm (ALF-funds) and The Sven Jerring Foundation. The funders had and will not have a role in study design, data collection and analysis, decision to publish, or preparation of the manuscript.

**Competing interests:** The authors have declared that no competing interests exist.

asymmetry) and number requiring brace treatment. Clinical follow-ups will be performed every six months and radiographs will be taken annually.

## Discussion

This study will compare effectiveness of an active self-corrective exercise strategy in mild idiopathic scoliosis with observation in terms of halting curve progression.

## Background

Idiopathic scoliosis is the most common spinal deformity in children and adolescents with an estimated prevalence of 3% [1]. About one tenth of the children with scoliosis develop a deformity that requires treatment with brace or surgery with the current treatment protocol. In the Swedish health care system, school screening programs account for at least 200,000 children screened for scoliosis per year. Children with a suspected scoliosis are usually referred to orthopedic specialists for evaluation. If a mild scoliosis exists, clinical observation during growth is the mainstay in Sweden. If progression occurs and the scoliosis requires treatment (Cobb $\geq$ 25 degrees), standard treatment in Sweden consists of bracing. The gold standard in treatment is a rigid thoracolumbosacral (TLSO) brace worn 18–20 hours or more per day, aiming to halt curve progression, although night-time brace is an alternative [2]. Previous literature on quality of life has suggested that brace treatment, being in many cases tough and cumbersome on adolescents, might have negative effects on mental health and studies have also shown that patients surgically treated for idiopathic scoliosis are to a larger extent satisfied with management [3–5].

Scoliosis specific exercises have in previous studies shown possible benefits in mild scoliosis curves and may be used as a method to prevent brace treatment, but the findings are not generally accepted [6–8]. The International Scientific Society on Scoliosis and Rehabilitation Treatment (SOSORT) recommends scoliosis-specific exercises (SSE) to be the first line of treatment in mild scoliosis to halt curve progression [9]. A recent review of SSE concluded that insufficient evidence exists supporting SSE in terms of improving scoliotic curves and warranted further high-quality evidence [10]. Furthermore, the lack of evaluation and follow-up until skeletal maturity [11] and comparison of scoliosis specific exercises with observation alone in mild scoliosis make valid conclusions difficult to draw [6]. Therefore, there is a need of a randomized controlled trial addressing these issues to determine the effectiveness of SSE for idiopathic scoliosis.

In a multicenter randomized trial, we aim to investigate the effectiveness of an active self-correction rehabilitation strategy with observation in adolescents with mild idiopathic scoliosis.

## The question at issue/hypothesis

We hypothesize that:

-Scoliosis specific exercises reduce the number of adolescents with scoliosis progression of more than 6 degrees, measured according to Cobb [12], on a standing frontal radiograph with a primary curve equal to or greater than 25 degrees compared to observation alone.

## Methods

### Study design

This study is a prospective randomized controlled trial. The protocol has been registered on ClinicalTrials.gov, identifier: NCT05138393. Currently, this project will be performed on two sites: Karolinska university hospital in Stockholm and Linköping university hospital in Linköping, Sweden. Further centers may be involved continuously. Individuals that meet inclusion criteria will be asked for participation. After informed consent, randomization will take place.

### Subjects

Patients referred to the participating clinics are consulted and examined by a health-care provider specialized in spinal disorders. The magnitude of the scoliosis in the frontal plane is assessed by measuring the Cobb angle [12]. Upon inclusion, full spine frontal and sagittal radiographs, pelvic and hand radiographs will be taken. Up to three months old radiographs will be accepted. Fig 1 illustrates the flow process for included patients and Fig 2 illustrates the recruitment process. The following inclusion criteria will be applied:

- Skeletally immature patients. Sander's score $\leq 4$ [13], Risser sign $< 2$ and no menarche for females.

- Age 9–15 years of age at inclusion.

- Primary curve of 15–24˚.

- Apex of the primary curve at T7 or caudal

- Scoliosis of idiopathic nature (i.e. no neuromuscular, congenital or syndromic origin).

### Interventions

#### Scoliosis specific exercises

The treatment will aim to teach the patients to become proficient in performing active self-correction of their scoliosis in the sagittal, coronal and horizontal planes. An additional step is combining the self-correction techniques in three dimensions (auto-correction in 3D) and stabilizing the corrected posture in accordance with SOSORT guidelines [9]. Patients will also be informed and educated in self-corrective behavior during task-oriented activities in daily living in order to enhance neuromotor control of the spine and limbs. Patients will have outpatient sessions once every two weeks the first 3 months and perform the exercises at home in 30-minutes sessions at least three times per week. Additional single bolus outpatient sessions may occur when extra education is required to master the program. Reinforcement of the intervention will be performed in conjunction with outpatient reassessment every six months. Patients are encouraged to continue with non-specific self-mediated physical activities of moderate intensity at least 60 minutes daily, for the entirety of the study. Compliance to the home exercise program will be monitored by using a mobile application (Physitrack, www. physitrack.com) where the patients record their sessions and can have contact with the research personnel. More detailed description of the scoliosis specific exercise intervention is provided in supplementary S1–S3 Files.

The head physiotherapists have a long and vast experience in managing and treating idiopathic scoliosis patients, both from a clinical and scientific perspective (together approximately 20 years of experience).

| | Study period | | | | | | | |
|---|---|---|---|---|---|---|---|---|
| | Enrollment | Allocation | Post-allocation | | | | | Close-out |
| Time point | Baseline | Baseline | 6 months | 12 months | 18 months | 24 months | Skeletal maturity | 2, 5 and 10 year follow-up |
| **Enrollment:** | | | | | | | | |
| Eligibility screen | x | | | | | | | |
| Informed consent | x | | | | | | | |
| Randomization | | x | | | | | | |
| **Interventions:** | | | | | | | | |
| Scoliosis-specific exercises | | | x | x | x | x | x | x |
| Observation | | | x | x | x | x | x | x |
| **Assessments:** | | | | | | | | |
| Full spine frontal radiograph | x | | x | | x | | x | x |
| Full spine sagittal radiograph | x | | x | | x | | x | x |
| Pelvic radiograph (Risser) | x | | | | | | | |
| Hand radiograph | x | | x | | x | | x | |
| Questionnaires | x | | x | x | x | x | x | x |
| Outpatient visit | x | | x | x | x | x | x | x |

**Fig 1. SPIRIT schedule of enrollment.**

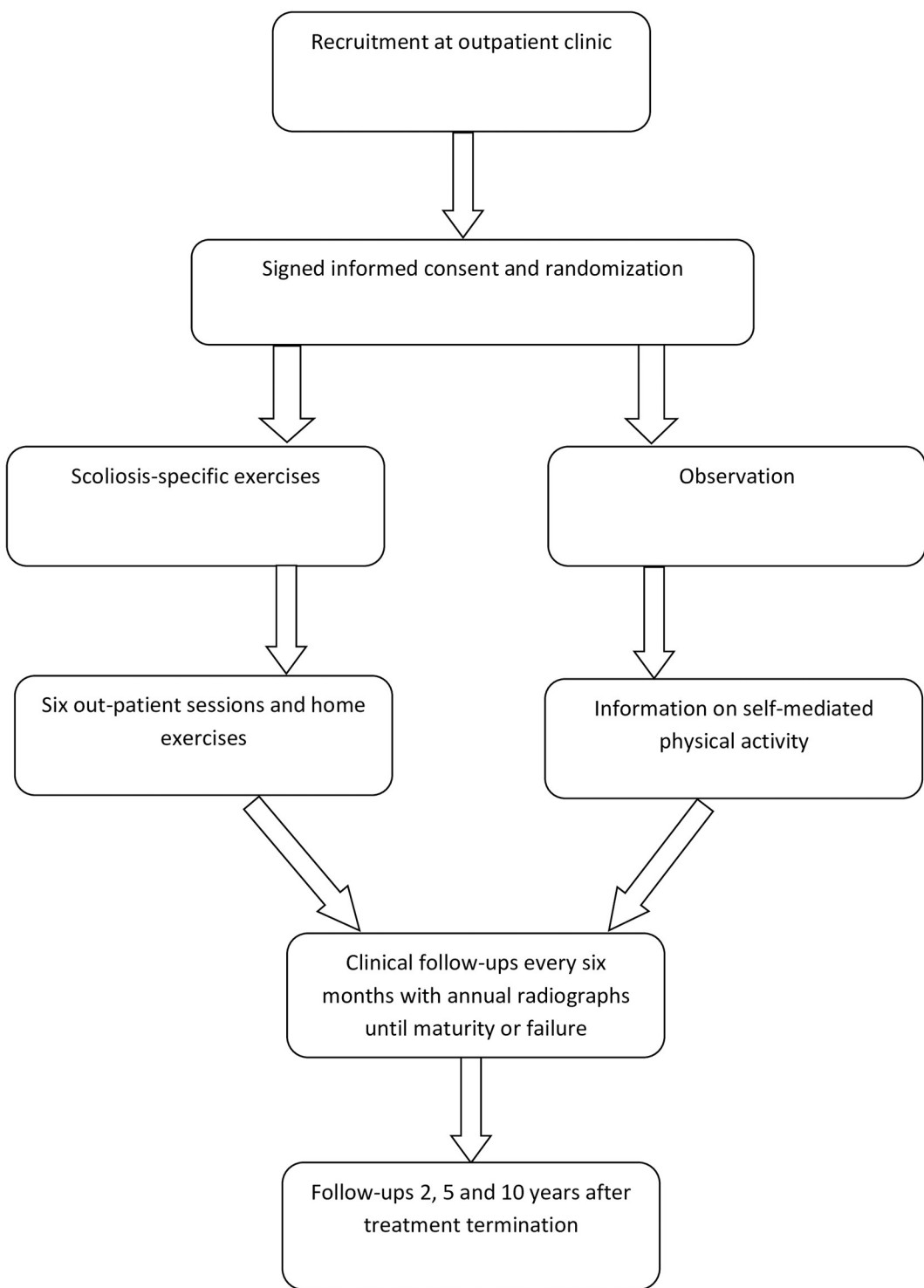

**Fig 2. Flow scheme of recruitment process.**

### Observational group (non-specific activity only)

Patients are encouraged to continue with non-specific self-mediated physical activities of moderate intensity at least 60 minutes daily, for the entirety of the study. If needed, the patients might be offered one outpatient session (remote or at the hospital) for information on self-mediated physical activity. Reinforcement of the intervention will be performed in conjunction with reassessment every six months.

### Non-participants

Patients not willing to undergo randomization will be treated in accordance with local procedures. Observation will be offered during growth and in the case of deterioration with a curve reaching at least 25 degrees in Cobb angle with remaining growth, fulltime TLSO brace treatment or night-time brace will be offered. The radiological and clinical result of these patients will be obtained from the regular orthopaedic files and radiographs taken.

### End of study

The study will be terminated when the participant reaches skeletal maturity, defined as less than 1.0 cm growth of body height in six months, or if the curve progresses more than 6 degrees, compared to the baseline radiograph, seen on two consecutive spinal standing radiographs. In case of curve progression and a Cobb angle surpassing 25 degrees, treatment will be offered with standard TLSO or nighttime brace to be worn 20 hours and 8 hours, respectively, per day. In the event that a Cobb angle surpasses 45 degrees with risk for further progression, patients will be offered surgical treatment. Follow-ups will be done in similar way also for those failing treatment. Post-treatment follow-up after endpoint is planned to occur at 2, 5 and 10 years prospectively including full spinal frontal and sagittal radiographs.

### Ethical permit

This study has been ethically approved by the Swedish Ethical Review Authority (approved 2021-04-20. Diary number: 2020–06502 and 2021–04632).

### Randomization procedure

After assessment, patients fulfilling the inclusion and exclusion criteria will be asked to give written and signed consent for participation. If the patient is aged less than 15 years, parental consent is also required. Patients who decline participation will be monitored as part of clinical routine.

Randomization will be done in a 1:1 ratio. Randomization will be performed using an online module available through the Swedish spine register (www.swespine.se). Randomization sequence have been prepared by an independent data manager at MedScinet (www.medscinet.com), the company that runs the register platform, and is unknown to the researchers.

### Blinding

Blinding of patients and clinicians is not possible. The statistician assessing the outcome will also be blinded to the type of treatment participants have had. In case of suspected progression during each follow-up including radiographs, two experts blinded for type of treatment will assess the radiographs to determine if progression has occurred.

## Outcome measurements

Primary outcome measure is progression of more than 6 degrees in Cobb angle seen on two consecutive radiographs from baseline and during radiographic follow-ups. Radiographs (hand and standing full-spine frontal and sagittal) will be taken every twelve months and clinical follow-ups will be performed every six months until skeletal maturity. The endpoint failure of treatment is defined as an increase of the Cobb angle of more than 6° from the time of the first radiograph on two consecutive examinations, considered as the minimal clinically important difference [14]. If progression is suspected, additional radiographs may be taken for verification, also at additional time points not being considered as regular follow-ups. The number of patients progressing to a curve requiring brace treatment will be recorded and compared in both groups. Indication for brace treatment is progression of more than 6 degrees together with a primary curve ≥ 25 degrees and remaining growth.

Secondary outcome measures recorded at baseline and every six months for the entirety of the study include angle of trunk rotation, as measured with Bunnell's scoliometer [15], SRS-22r and EQ-5D-Y quality of life questionnaires, VAS-pain, the International Physical Activity Questionnaire (IPAQ-SF) short form, Spinal Appearance Questionnaire (SAQ) and number requiring brace treatment.

In line with the capability, opportunity, motivation, and behaviour change model (COM-B) [16], to gain knowledge of behavioural outcomes, all study participants will answer three additional questions during each six-month follow-up; 1) the grade to which you feel that you have completed the treatment (adherence), 2) the grade to which you are motivated to carry out the treatment (motivation) and 3) how confident are you in your own capability to perform the treatment (capability). Additionally, the healthcare provider answers the question "to what grade the patient has adhered to treatment plan" (patient adherence). These questions are rated on a scale from best "very sure" (1 point) to worst "not at all" (4 points).

## Data integrity

Data will be entered by the participant directly in an online module, or by paper format and then entered by the research staff.

## Data analysis

Data from the trial will be compared based on the 'intention to treat' principle. An intention to treat (ITT) analysis means that all patients, regardless of non-compliance, loss to follow-up or drop-out, remain in the analysis of the group to which they were randomized [17]. Depending on the mechanisms for missing data, the most relevant imputation method considering missing at random or not missing at random will be used in the ITT analysis [18]. A causal survival analysis measuring complier average causal effects (CACE), where standardization of baseline covariates (age, sex, Risser sign, major curve) and inverse probability weighting of post-randomization treatment adherence data is used in a per protocol analysis, to be contrasted with ITT results [19]. With basis in the COM-B model, patient reported capability and motivation can be hypothesized as potential mediators of treatment effect on the outcome variables. Fig 3 summarizes these causal inferences in a direct acyclic graph (DAG) model [20]. If a patient fails treatment and is offered another treatment, such as brace or surgery, the Cobb angle at the moment of commencing this treatment will be considered in the secondary outcome analysis of Cobb angle. Categorical parameters will be compared by the Chi-square test. Continuous and discrete parameters will be measured using parametric or non-parametric tests (depending on skewness) for group comparisons. Individual variables will be tested for their association with treatment effect by adding a predictor × treatment group interaction term to

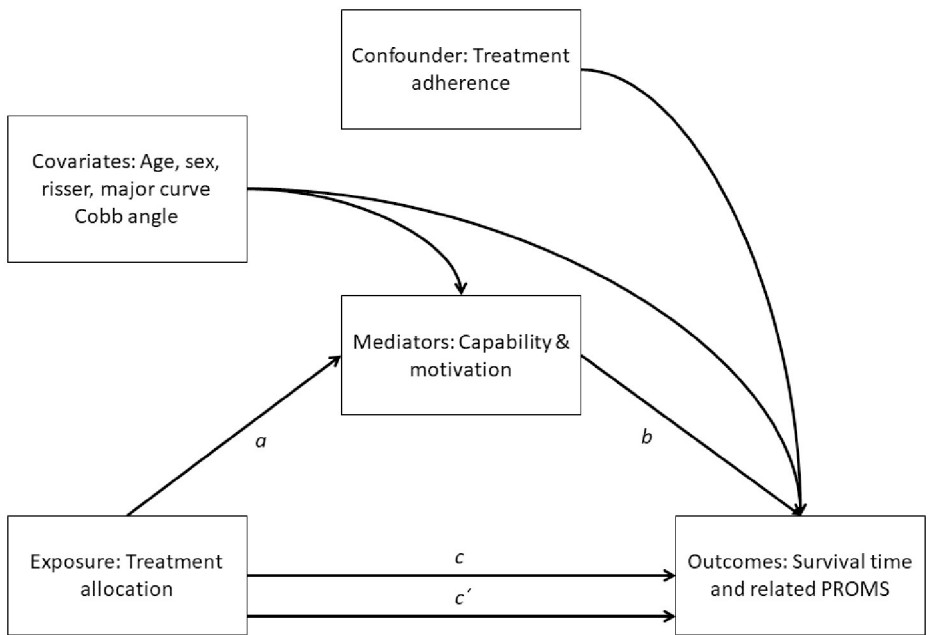

**Fig 3. Directed acyclic graph of hypothesized mechanisms in single mediator models in the PREMISCOPRO trial and the estimated averaged effects standardized for baseline covariates and inverse probability weighted for post-randomization treatment adherence confounding.** The indirect effect (*ab*-product) is the average intervention effect through the mediator. *a*, a-path (the intervention-mediator effect); *b*, b-path (the mediator-outcome effect); *c*, c-path (the total effect of the intervention on the outcome, without accounting for potential mediator); *c′* (the direct effect of the intervention on the outcome, that works through all other mechanisms excluding the selected potential mediator).

a regression equation. Additionally, Kaplan-Meier survival analyses will be used to display the probability of more than 6˚ Cobb progression over time. Cox regression or pooled logistic regression will be used to provide survival hazard ratios. The results will be reported according to Consolidated Standards of Reporting Trials (CONSORT) [21].

## Power analysis

The end point failure of treatment is defined as an increase of the Cobb angle of more than 6˚ on two consecutive x-rays, when compared to the x-ray performed at time of inclusion. Based on a hypothesized failure rate of 10% in the scoliosis specific exercise group and 39% in the observation group [6], with a significance level of 5% and a power of 80% and consideration for dropout of up to 20% and an additional adding of five individuals per group, an estimated number of 45 individuals in each group is required.

## Discussion

This multicenter study on exercise treatment with an active self-corrective strategy for mild idiopathic scoliosis will determine effectives compared with observation alone. The project includes key methodological features in order to minimize bias in clinical trials such as true randomization, specification of eligibility criteria, blinding and intention-to-treat analysis. The methodological framework of this project is similar to a successfully managed randomized, but yet unpublished, trial from our group [22].

Our choice of outcomes and inclusion criteria is in line with recommendations for clinical trials studying idiopathic scoliosis. Scoliosis-specific exercises, as of today, are not routinely

prescribed for scoliosis patients in Swedish healthcare. The mainstay for mild scoliosis is observation during growth. Our choice to use an active self-correction strategy is based on an impartial approach to synthesizing a broad intervention covering principles outlined in scoliosis specific exercise literature. This approach has been used successfully in randomized trials previously [6, 22]. We believe that adolescents who can actively manage their scoliosis, with proper education and information about the condition, will have better patient-reported outcomes in terms of increased self-esteem (mental health and self-image) and self-empowerment (capability and motivation).

## Supporting information

**S1 File. Specifications for interventional group.**
(DOCX)

**S2 File. Behavioural changes and increasing compliance.**
(DOCX)

**S3 File. The TIDieR (Template for Intervention Description and Replication) checklist.**
(DOCX)

**S4 File. Project plan submitted to the Swedish ethical review authority.**
(PDF)

**S5 File. WHO trial registration.**
(DOCX)

**S6 File. SPIRIT checklist.**
(DOC)

## Author Contributions

**Conceptualization:** Elias Diarbakerli, Allan Abbott, Paul Gerdhem.

**Data curation:** Elias Diarbakerli, Allan Abbott, Paul Gerdhem.

**Formal analysis:** Elias Diarbakerli, Allan Abbott.

**Funding acquisition:** Elias Diarbakerli.

**Investigation:** Elias Diarbakerli, Allan Abbott, Paul Gerdhem.

**Methodology:** Elias Diarbakerli, Allan Abbott, Paul Gerdhem.

**Project administration:** Elias Diarbakerli, Allan Abbott, Paul Gerdhem.

**Resources:** Elias Diarbakerli.

**Software:** Elias Diarbakerli.

**Writing – original draft:** Elias Diarbakerli.

**Writing – review & editing:** Elias Diarbakerli, Allan Abbott, Paul Gerdhem.

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
