## [Decision Letter · Decision Letter 0]

6 Sep 2022

PONE-D-22-18687PREventing Mild Idiopathic SCOliosis PROgression (PREMISCOPRO): a protocol for a Randomized Controlled Trial Comparing Scoliosis-specific Exercises with Observation in Mild Idiopathic ScoliosisPLOS ONE

Dear Dr. Diarbakerli,

Thank you for submitting your manuscript to PLOS ONE. After careful consideration, we feel that it has merit but does not fully meet PLOS ONE’s publication criteria as it currently stands. Therefore, we invite you to submit a revised version of the manuscript that addresses the points raised during the review process.

ACADEMIC EDITOR:Criteria for included age, 10 years and 14 years to be revisited.Consider the measurement error of 5 degree while using Cobb's angle.Is sample size based on stratificationadherence measured and compliance, not to be confused. To be made clear Please ensure that your decision is justified on PLOS ONE’s publication criteria and not, for example, on novelty or perceived impact. For Lab, Study and Registered Report Protocols: These article types are not expected to include results but may include pilot data. 

We look forward to receiving your revised manuscript.

Kind regards,

Asir John Samuel, Ph.D.

Academic Editor

PLOS ONE

Journal Requirements:

3. Please ensure that you refer to Figures 1 and 2 in your text as, if accepted, production will need this reference to link the reader to the figure.

4. We note that the original protocol that you have uploaded as a Supporting Information file contains an institutional logo. As this logo is likely copyrighted, we ask that you please remove it from this file and upload an updated version upon resubmission.

Additional Editor Comments:

Congratulate for your wonderful work. However, modifications as suggested by the reviewers to be incorporated.

Reviewers' comments:

Reviewer's Responses to Questions

**Comments to the Author**

1. Does the manuscript provide a valid rationale for the proposed study, with clearly identified and justified research questions?

Reviewer #1: Yes

Reviewer #2: Yes

Reviewer #3: Yes

2. Is the protocol technically sound and planned in a manner that will lead to a meaningful outcome and allow testing the stated hypotheses?

Reviewer #1: Yes

Reviewer #2: Yes

Reviewer #3: Yes

3. Is the methodology feasible and described in sufficient detail to allow the work to be replicable?

Reviewer #1: Yes

Reviewer #2: Yes

Reviewer #3: Yes

4. Have the authors described where all data underlying the findings will be made available when the study is complete?

Reviewer #1: Yes

Reviewer #2: Yes

Reviewer #3: No

5. Is the manuscript presented in an intelligible fashion and written in standard English?

Reviewer #1: Yes

Reviewer #2: Yes

Reviewer #3: Yes

6. Review Comments to the Author

You may also provide optional suggestions and comments to authors that they might find helpful in planning their study.

Reviewer #1: Congratulations to the author/s for planning the research in a very important area. The protocol is well written and has followed the randomized controlled trial protocol guidelines appropriately. However, few minor revisions should be done before considering for acceptance as mentioned below:

1) Intervention image with proper labelling can be added for more clearance about the technique.

2) Reference manager should be followed for a citation and few of the references in the list are incomplete.

Reviewer #2: This protocol address a very relevant question about the role of exercises in preventing scoliosi progression. This is a crucial point in current knowledge about scoliosis treatment.

I have some suggestions about the inclusion criteria.

I would exclude patients below 10 years, so to include patients in a higher risk phase and only AIS. I would exclude also patients older than 14, since we can expect growth to be very close to the end. I understand that using the Sanders and Risser scores would help about this, but I would be more selective.

Since curves up to 24° will be included, and the measurement error of Cobb is 5°, I would wait 30° to brace patients. Otherwise there is the theoretical risk to brace a patient that changes from 24 to 25 or 26 which is not actually a changed and showing and acceptable condition.

I would also add other secondary outcomes: the number of improved patients (6° or more), the number of patients who completed growth below 30°, which is a main goal according to SOSORT guidelines. I think that reducing the progression rate even without totally stop it would be successful for the patients in case the threshold of 30° is not reached.

Please describe the protocol of exercises applied. Will it be SEAS, Schroth or other?

Blinding: I suggest to have a blind evaluation by an expert of the radiographies to doublecheck the treating physician's evaluation. This would make the paper much stronger.

Reviewer #3: This is an interesting study looking at Scoliosis specific exercises to reduce the number of adolescents with scoliosis progression of more than 6 degrees.

Some minor comments worth addressing.

1. Although the sample size is based on 90 people, did the authors consider stratification of perhaps for primary curve and age, as these were adjusted for in the models?

2. In the data analysis section, can the authors include that results will be present according to CONSORT.

3. A CACE analysis will be used, can the authors define compliance here.

4. In the data analysis, it would help readability to split primary and secondary outcome analysis. Also some outcomes are collected at repeated time points, does the analysis take this into account?

5. Can the authors also state the results will be presented as treatment effect, confidence interval together with associated p-values.

6. How is treatment adherence measured, not to be confused with compliance, or can the authors really clarify this.

7. PLOS authors have the option to publish the peer review history of their article (what does this mean?). If published, this will include your full peer review and any attached files.

Reviewer #1: No

Reviewer #2: No

Reviewer #3: No

---

## [Author Response · Author response to Decision Letter 0]

18 Oct 2022

Respond to reviewers

Academic editor

Q: Criteria for included age, 10 years and 14 years to be revisited.

A: Thank you for the suggestion. The age span 9-15 years was chosen based on that adolescents, boys especially, may have a lot of remaining growth also after the age of 14. The lower threshold 9 was applied to be able to include children surpassing the age of 9 but not yet reached 10. The study is ongoing with a published protocol in clinicaltrials.gov. No changes have been made to the manuscript.

Q: Consider the measurement error of 5 degree while using Cobb's angle.

A: We have chosen 6 degrees in line with our previous publications (Diarbakerli et al, Plos one 2021 and Dufvenberg et al, J Clin Med 2021) to be able to make future comparisons between cohorts in these studies. This study is ongoing with a published protocol in clinicaltrials.gov. No changes have been made to the manuscript.

Q: Is sample size based on stratification adherence measured and compliance, not to be confused. To be made clear

A: We thank the editor for the question. We do not quite understand it completely but it is stated in the power calculation’s section that sample size is based on previous reported success rates. If the editor could elaborate on the question further we will be happy to answer it further.

Reviewer 1

Q: Intervention image with proper labelling can be added for more clearance about the technique.

A: We thank the reviewer for this suggestion. We have tried to describe the intervention in detail in the supplementary files. The core of the intervention is in line with previous literature and guidelines from SOSORT, namely that individual active correction in all planes is performed by the patients. Interventions can therefore differ for patients based on clinical characteristics and curve localization. No changes have been made to the manuscript.

Q: Reference manager should be followed for a citation and few of the references in the list are incomplete.

A: We thank the reviewer for these observations. We have updated the references accordingly. 

Reviewer 2

Q: I would exclude patients below 10 years, so to include patients in a higher risk phase and only AIS. I would exclude also patients older than 14, since we can expect growth to be very close to the end. I understand that using the Sanders and Risser scores would help about this, but I would be more selective. Since curves up to 24° will be included, and the measurement error of Cobb is 5°, I would wait 30° to brace patients. Otherwise there is the theoretical risk to brace a patient that changes from 24 to 25 or 26 which is not actually a changed and showing and acceptable condition. I would also add other secondary outcomes: the number of improved patients (6° or more), the number of patients who completed growth below 30°, which is a main goal according to SOSORT guidelines. I think that reducing the progression rate even without totally stop it would be successful for the patients in case the threshold of 30° is not reached.

Please describe the protocol of exercises applied. Will it be SEAS, Schroth or other?

Blinding: I suggest to have a blind evaluation by an expert of the radiographies to doublecheck the treating physician's evaluation. This would make the paper much stronger.

A: We thank the reviewer for the questions and suggestions which we to a large extent agree upon. Regarding age, 9-15 years was chosen based on that adolescents, boys especially, may have a lot of remaining growth also after the age of 14. The lower threshold 9 was applied to be able to include children surpassing the age of 9 but not yet reached 10. These are also the threshold we have used in our previous studies. 

In terms of brace initiation; it is stated in the manuscript that brace treatment will be offered to patients progressing > 6 degrees Cobb AND surpassing 25 degrees. A patient with a curve of 24 degrees with a follow-up radiograph showing a few degrees change will therefore not be equal to progression. 

In terms of secondary outcomes, we have included a bunch both in terms of PROM data but also number progressing to the need of brace treatment which we believe is most interesting from a clinical perspective, that is, can we with SSE save more patients from ever needing a brace? 

The methods used is a mix of the available literature and is mainly based on an active self-corrective strategy. As described in the manuscript and supplementary files, there is an exercise part (active self-correction) but also an educational part including activity in daily tasks where the child will learn how to move away from convexity to concavity, general loading of the spine, ergonomics etc. 

When it comes to radiographs and evaluation of them, there will always be at least two persons assessing radiographs. If they can not reach consensus, a third expert will also be consulted. This is described in page 7 under the section outcome measurements. No changes have been made to the manuscript.

Reviewer 3

Q: Although the sample size is based on 90 people, did the authors consider stratification of perhaps for primary curve and age, as these were adjusted for in the models?

A: We thank the reviewer for the question. The power calculation is based on previous studies and success rates, with consideration for drop outs. No changes have been made to the manuscript.

Q: In the data analysis section, can the authors include that results will be present according to CONSORT.

A: We thank the reviewer for this suggestion. This has been added as the last sentence under the data analysis section, page 9.

Q: A CACE analysis will be used, can the authors define compliance here.

A: Information on compliance and recommended dose of exercises are described in page 5. Data on compliance will be extracted for the mobile application and also from surveys collected at each 6 month follow-up, detailed in pages 7-8.

Q: In the data analysis, it would help readability to split primary and secondary outcome analysis. Also some outcomes are collected at repeated time points, does the analysis take this into account?

A: The section data analysis discusses and describes how data will be handled. We believe the reviewer is referring to the section called outcome measurements where the primary and secondary outcome measurements are described in different sections. No changes have been made to the manuscript.

Q: Can the authors also state the results will be presented as treatment effect, confidence interval together with associated p-values.

A: We thank the reviewer for this suggestion. Since this is a study protocol, we have described how data will be handled thoroughly, including how for example categorical and continuous data will be presented. We refer to the data analysis section. No changes have been made to the manuscript. 

Q: How is treatment adherence measured, not to be confused with compliance, or can the authors really clarify this

A: We thank the reviewer for this question. As stated in page 8, all study participants will answer three additional questions during each six-month follow-up; 1) the grade to which you feel that you have completed the treatment (adherence), 2) the grade to which you are motivated to carry out the treatment (motivation) and 3) how confident are you in your own capability to perform the treatment (capability). Additionally, the healthcare provider answers the question “to what grade the patient has adhered to treatment plan” (patient adherence). These questions are rated on a scale from best “very sure” (1 point) to worst “not at all” (4 points). No additional changes have been made to the manuscript.

---

## [Decision Letter · Decision Letter 1]

8 Dec 2022

PONE-D-22-18687R1PREventing Mild Idiopathic SCOliosis PROgression (PREMISCOPRO): a protocol for a Randomized Controlled Trial Comparing Scoliosis-specific Exercises with Observation in Mild Idiopathic ScoliosisPLOS ONE

Dear Dr. Diarbakerli,

Thank you for submitting your manuscript to PLOS ONE. After careful consideration, we feel that it has merit but does not fully meet PLOS ONE’s publication criteria as it currently stands. Therefore, we invite you to submit a revised version of the manuscript that addresses the points raised during the review process.

ACADEMIC EDITOR: Please insert comments here and delete this placeholder text when finished. Be sure to:

The issues I raised previously appear to have been addressed. However, minor revision need to be made as suggested by one of the reviewer. 

We look forward to receiving your revised manuscript.

Kind regards,

Asir John Samuel, Ph.D.

Academic Editor

PLOS ONE

Journal Requirements:

Additional Editor Comments:

The issues raised previously appear to have been addressed.

However, minor revision need to be made as suggested by one of the reviewer.

The comments are as follows,

Again, I think this is an overall fundamental and well designed study. Nevertheless, there are some points that are questionable since are not consistent with current literature.

Including patients younger than 10 means mixing JIS and AIS, makes results more confuse will take longer to reach the final outcomes increasing costs. Stating that this consistent with your previous publications is weak, since the rest of the world differs from this.

There are 2 protocols with somehow solid data showing positive results in the field: Schroth and SEAS. To perform one of them the PTs need a specific certification. The authors state they will do something different coming from the literature and taking ideas from different approaches. What is the experience of the PTs administrating this treatment to patients? Do they have any certification (Schroth, SEAS or other)? How can readers know about the quality of the treatment? Please give more details.

Also, the protocol is not clear about how many training sessions will be administered after the initial phase. Again, Schroth and SEAS have a quite clear protocol to ensure quality and a continuous contact with the therapist, since if patients work too much by themselves the precision in the Active self correction tend to be less precise, and this can affect the final results.

I appreciate the fact that radiographs will be assessed in blind. I would add a sentence about this also in the pragraph "blinding".

Reviewers' comments:

Reviewer's Responses to Questions

**Comments to the Author**

1. Does the manuscript provide a valid rationale for the proposed study, with clearly identified and justified research questions?

Reviewer #1: Yes

Reviewer #2: Yes

Reviewer #3: Yes

2. Is the protocol technically sound and planned in a manner that will lead to a meaningful outcome and allow testing the stated hypotheses?

Reviewer #1: Yes

Reviewer #2: Partly

Reviewer #3: Yes

3. Is the methodology feasible and described in sufficient detail to allow the work to be replicable?

Reviewer #1: Yes

Reviewer #2: Yes

Reviewer #3: Yes

4. Have the authors described where all data underlying the findings will be made available when the study is complete?

Reviewer #1: Yes

Reviewer #2: Yes

Reviewer #3: No

5. Is the manuscript presented in an intelligible fashion and written in standard English?

Reviewer #1: Yes

Reviewer #2: Yes

Reviewer #3: Yes

6. Review Comments to the Author

You may also provide optional suggestions and comments to authors that they might find helpful in planning their study.

Reviewer #1: All the queries raised are answered successfully by the authors. However, in the future studies more meticulously the limitations of the study should be addressed

Reviewer #2: Again, I think this is an overall fundamental and well designed study. Nevertheless, there are some points that are questionable since are not consistent with current literature.

Including patients younger than 10 means mixing JIS and AIS, makes results more confuse will take longer to reach the final outcomes increasing costs. Stating that this consistent with your previous publications is weak, since the rest of the world differs from this.

There are 2 protocols with somehow solid data showing positive results in the field: Schroth and SEAS. To perform one of them the PTs need a specific certification. The authors state they will do something different coming from the literature and taking ideas from different approaches. What is the experience of the PTs administrating this treatment to patients? Do they have any certification (Schroth, SEAS or other)? How can readers know about the quality of the treatment? Please give more details.

Also, the protocol is not clear about how many training sessions will be administered after the initial phase. Again, Schroth and SEAS have a quite clear protocol to ensure quality and a continuous contact with the therapist, since if patients work too much by themselves the precision in the Active self correction tend to be less precise, and this can affect the final results.

I appreciate the fact that radiographs will be assessed in blind. I would add a sentence about this also in the pragraph "blinding".

Reviewer #3: All previous comments have been checked and the authors have addressed all comments. I have no further comments.

7. PLOS authors have the option to publish the peer review history of their article (what does this mean?). If published, this will include your full peer review and any attached files.

Reviewer #1: No

Reviewer #2: No

Reviewer #3: No

---

## [Author Response · Author response to Decision Letter 1]

20 Dec 2022

Reviewer 1

Q: All the queries raised are answered successfully by the authors. However, in the future studies more meticulously the limitations of the study should be addressed.

A: We thank the reviewer for the previous valuable comments and suggestions.

Reviewer 2

Q: Again, I think this is an overall fundamental and well designed study. Nevertheless, there are some points that are questionable since are not consistent with current literature.

Including patients younger than 10 means mixing JIS and AIS, makes results more confuse will take longer to reach the final outcomes increasing costs. Stating that this consistent with your previous publications is weak, since the rest of the world differs from this.

A: We thank the reviewer for this suggestion. As previously mentioned, the lower limit of age 9 years was applied in order to have the possibility to include patients being just below the age of 10 (i.e. 9 years and 11 months) and also to be in line with our previous studies in order to facilitate meta-analyses in the future. Since we have quiet strict criteria in terms of maturity (no menarche, Risser < 2 and Sanders 4 or below) we believe that many patients between ages 9-10 will be excluded if the lower border is 10 years strictly. It is true that some patients with JIS might be included but we believe they will be in clear minority and we do not believe this will affect the studies result. 

Like previously mentioned; this is an ongoing study with a published protocol in clinicaltrials.gov. Changes in inclusion criteria is therefore not a possibility. 

Q: There are 2 protocols with somehow solid data showing positive results in the field: Schroth and SEAS. To perform one of them the PTs need a specific certification. The authors state they will do something different coming from the literature and taking ideas from different approaches. What is the experience of the PTs administrating this treatment to patients? Do they have any certification (Schroth, SEAS or other)? How can readers know about the quality of the treatment? Please give more details.

Also, the protocol is not clear about how many training sessions will be administered after the initial phase. Again, Schroth and SEAS have a quite clear protocol to ensure quality and a continuous contact with the therapist, since if patients work too much by themselves the precision in the Active self correction tend to be less precise, and this can affect the final results.

A: We thank the reviewer for these important issues that are raised.

As we know, there is no level 1 evidence favoring scoliosis-specific exercises. This is not a generally accepted form of treatment for patients with idiopathic scoliosis. We do not intend to label the interventions as “Schroth”, “SEAS”, “FITS”, “side-shift therapy” etc and therefore a specific “certification” is not required for this study. SOSORT has stated in their latest guidelines (Negrini et al 2016) that the studies available are “generally of poor quality”. The guidelines state further:

“SOSORT experts agree that PSSE should consist of the following:

Auto-correction in 3D

Training in activities of daily living (ADL)

Stabilizing the corrected posture

Patient education”

All of these variables are addressed in the current study. Additionally, the study by Monticone et al from 2014 used a similar approach as this study protocol, not being tied to a specific “school” but to target the scoliosis patients individually based on above parameters agreed upon by SOSORT. 

The senior physiotherapists involved in this study have long and vast experience in treating and managing scoliosis patients, both in terms of clinical and scientific point of view where the most senior PTs hold PhDs. The PTs have been participants and presenters in multiple SRS and SOSORT meetings. We have added the following sentence under the paragraph “interventions”: The head physiotherapists have a long and vast experience in managing and treating idiopathic scoliosis patients, both from a clinical and scientific perspective (together approximately 20 years of experience).

In the manuscript, it is stated on pages 5 and 6 (and supplementary files) under the headline “interventions” that patients in the interventional group will have outpatient sessions on 6 occasions during the first 3 months and thereafter will continue with their program according to prescription with follow-ups every six months. Above this, they have direct contact with research personnel via the mobile application described and are entitled to either digital or physical sessions whenever needed during the entire study period. In terms of frequency of exercises, SOSORT guidelines state “The frequency of therapeutic sessions varies from twice to 7 days a week depending on the complexity of the techniques, motivation and the ability of the patient to carry out the treatment. Long-term outpatient physiotherapy sessions usually take place two to four times a week if the patient is willing to cooperate fully”. Looking at the context for this study protocol, treatment is carried out at a large university hospital. The decision having outpatient sessions intense for the first three months is balanced against the willing and motivation for families to travel, leave school etc for a generally benign condition they are barely bothered by. Therefore, during the first three months, we aim to educate and make sure the child has a sense of control of his/her situation and treatment. If we identify difficulties, there are no restrictions for multiple outpatient sessions after the first three months. This is specified on page 5, last section.

Q: I appreciate the fact that radiographs will be assessed in blind. I would add a sentence about this also in the pragraph "blinding".

A: We thank the reviewer for this comment. It is stated in the manuscript that the statistician assessing outcome will be blinded for type of treatment. The following sentence have been added under the paragraph “blinding”: In case of suspected progression during each follow-up including radiographs, two experts blinded for type of treatment will assess the radiographs to determine if progression has occurred.

Reviewer 3

Q: All previous comments have been checked and the authors have addressed all comments. I have no further comments.

A: We thank the reviewer for the previous valuable comments and suggestions.

---

## [Editor Report · Decision Letter 2]

19 Apr 2023

PREventing Mild Idiopathic SCOliosis PROgression (PREMISCOPRO): a protocol for a Randomized Controlled Trial Comparing Scoliosis-specific Exercises with Observation in Mild Idiopathic Scoliosis

PONE-D-22-18687R2

Dear Dr. Diarbakerli,

We’re pleased to inform you that your manuscript has been judged scientifically suitable for publication and will be formally accepted for publication once it meets all outstanding technical requirements.

Kind regards,

Felicity Hey

Staff

PLOS ONE

On behalf of 

Dr. Asir John Samuel

Academic Editor

---

## [Editor Report · Acceptance letter]

27 Apr 2023

PONE-D-22-18687R2 

PREventing Mild Idiopathic SCOliosis PROgression (PREMISCOPRO): a protocol for a Randomized Controlled Trial Comparing Scoliosis-specific Exercises with Observation in Mild Idiopathic Scoliosis 

Dear Dr. Diarbakerli:

I'm pleased to inform you that your manuscript has been deemed suitable for publication in PLOS ONE. Congratulations! Your manuscript is now with our production department. 

Kind regards, 

on behalf of

Dr. Asir John Samuel 

Academic Editor

PLOS ONE